# Human Umbilical Cord: Information Mine in Sex-Specific Medicine

**DOI:** 10.3390/life11010052

**Published:** 2021-01-13

**Authors:** Ilaria Campesi, Flavia Franconi, Andrea Montella, Salvatore Dessole, Giampiero Capobianco

**Affiliations:** 1Dipartimento di Scienze Biomediche, Università degli Studi di Sassari, 07100 Sassari, Italy; montella@uniss.it; 2Laboratorio Nazionale sulla Farmacologia e Medicina di Genere, Istituto Nazionale Biostrutture Biosistemi, 07100 Sassari, Italy; franconi.flavia@gmail.com; 3Unità Operativa di Genetica e Biologia dello Sviluppo, Azienda Ospedaliero Universitaria di Sassari, 07100 Sassari, Italy; 4Dipartimento di Scienze Mediche, Chirurgiche e Sperimentali, Clinica Ostetrica e Ginecologica, Università degli Studi di Sassari, 07100 Sassari, Italy; dessole@uniss.it (S.D.); capobia@uniss.it (G.C.)

**Keywords:** umbilical cord, sex differences, fetal programming, preclinical research

## Abstract

Biological differences between sexes should be considered in all stages of research, as sexual dimorphism starts in utero leading to sex-specific fetal programming. In numerous biomedical fields, there is still a lack of stratification by sex despite primary cultured cells retaining memory of the sex and of the donor. The sex of donors in biological research must be known because variations in cells and cellular components can be used as endpoints, biomarkers and/or targets of pharmacological studies. This selective review focuses on the current findings regarding sex differences observed in the umbilical cord, a widely used source of research samples, both in the blood and in the circulating cells, as well as in the different cellular models obtainable from it. Moreover, an overview on sex differences in fetal programming is reported. As it emerges that the sex variable is still often forgotten in experimental models, we suggest that it should be mandatory to adopt sex-oriented research, because only awareness of these issues can lead to innovative research.

## 1. Introduction

Several diseases (cardiovascular, inflammatory and autoimmune diseases, diabetes mellitus, cancer, depression and brain disorders, and infections) are affected by sex differences in their diffusion, progression, and treatment [1,2,3]. Therefore, biological differences between sexes should be taken into account in all stages of research, from the pre-analytic conditions to genetics, epigenetics [4,5], developmental biology, biochemistry, physiology, pharmacology, toxicology, and epidemiology as well as social sciences, using all new technologies including omics [4].

Sexual dimorphism starts in utero and seems to occur at a pre-gonadal stage [6,7,8]. Moreover, fetal programming, which predispose developing organism to increased risk for future diseases appears to be strongly influenced by the fetus sex [9,10].

Although sex-specific differences depend on animal species and strains, in most biomedical research, almost all cellular studies [11] do not differentiate between genetic male or female cells, and a high proportion of preclinical studies (68–76%) use only males or do not report the sex of the animals [12,13]. Sex, in fact, should be considered in all cell studies, as it is now evident that different primary cells from males and females behave differently [14,15,16,17,18,19,20,21,22]. The stratification of cells according to the sex of donors become fundamental, because organelles and cells have memory of their sex [15,16,17,18,19,23] and differences encountered in cellular compartments can be used as end points, biomarkers and/or targets of pharmacological studies [2].

The human umbilical cord has no particular ethical impediments, is non-tumorigenic, and less immunogenic, representing an advantageous experimental source over other cell sources [24,25]. Moreover, it seems a good experimental model for studying and understanding sex differences that characterize the cardiovascular system [1]. This selective review focuses on the current findings regarding sex differences observed in the umbilical cord, both in the blood and in the circulating cells, as well as in the different cellular models obtainable from it (Figure 1).

## 2. Strategy Search

Literature analysis was performed using PubMed and Google as research tools using the following key-words and their combination: gender differences, sex differences, males, females, biomarkers, umbilical cord, cord blood, placenta, serum, plasma, progenitor cells, mononuclear cells, human umbilical vein endothelial cells (HUVECs), human umbilical artery endothelial cells (HUAECs), human umbilical artery smooth muscle cells (HUASMCs), Wharton Jelly (WJ), mesenchymal stem cells (MSCs), and fetal programming.

## 3. Sex Differences in Fetal Growth and Placenta

Sex determination occurs in two stages: in the first, sex chromosomes guide the differentiation of bipotential gonadal crests in the testis or ovary [26]. In the second stage, gonadal sex hormones drive the creation of a number of anatomical and physiological features known as phenotypic sex [27]. The inactivation of the X chromosome is an important epigenetic process that occurs in mammalian females to correct the imbalance of the X chromosome genes between the sexes, a phenomenon that determines a transcriptional silencing aimed at obtaining an equal gene dosage; however, the silencing is often not complete [28,29]. In fact, 15–30% of human X-linked genes may escape this process, creating protein level differences between male and female individuals and making females more susceptible to certain diseases than males (such as autoimmune diseases), or protect them from other conditions such as cancer [29,30,31,32].

The upregulation/downregulation of transcription factors should initiate the differences in development, and the observation that sex-determining region Y (SRY) factor induces cell proliferation in the fetal mouse gonads [33] further emphasizes the importance of differential growth in sex determination and differentiation. SRY factor is an additional growth promoter gene that allows the XY embryo to differentiate into the female hormonal environment of the uterus. It is noteworthy that XX mice carrying a SRY-box transcription factor 9 (SOX9) transgene were found to develop as males [34], and that the importance of transcription factors is also reported by Colvin et al. [35] which showed that most XY mice lacking fibroblast growth factor 9 (FGF9) developed as females.

Female fetuses had smaller cord areas and less WJ than male fetuses, and male and female fetuses develop their length of the cord and relative placental weight differently from each other under the influence of umbilical ring constriction. Sex-specific differences in fetus growth appear early in the pregnancy and have long been recognized [36]. Cell division is more rapid in male embryos than in female ones [37], and male fetuses growth seems to be greater than the female ones. Crown-rump length and biparietal diameter (BPD) in human male fetuses are, on average, larger than in females from the first trimester until 15th week of gestation [38,39,40]. Moore described significant differences in head growth trajectories between male and female fetuses, showing that the head and abdominal circumferences were higher in male fetuses starting in the second trimester [41]. Moreover, Galjaard and colleagues observed that BPD and head circumference were significantly larger in males than in females from 20 weeks of gestation onwards [42]. These observations suggest that males may be both more responsive to growth promoting influences, and more susceptible to supply disturbances. Moreover, male fetuses are more active than female fetuses [43,44,45].

Anti-Müllerian hormone (AMH), a member of the transforming growth factor-β (TGF-β) super-family, is produced by Sertoli cells from the onset of testicular differentiation and by granulosa cells after birth in mammals [46], which play a role in sexual differentiation and recruitment. It binds to a serine-threonine kinase receptor complex consisting of ligand-specific type II receptors (AMHRII), recruiting and phosphorylating more general type I receptors also known as activin receptor-like protein kinases (ALKs). It plays key roles in the regression of the Müllerian duct in the male embryo. Indeed, gonadal sex steroids are necessary for the sexual differentiation of the fetus and for sexual maturation during prepubertal to pubertal age. In the stages of sexual maturation Sertoli and granulosa cells develop from a common precursor: the somatic cells expressing SRY differentiate into Sertoli cells and Leyding cells (testis), while somatic cells in which SRY is not expressed differentiate into granulosa cells and theca cells (ovary). The two groups of somatic cells subsequently acquire sex-specific functions, including the synthesis of sex steroids. AMH is at the crossroads of sexual determination and differentiation, and, after SRY and SOX9, it is the first product identified that characterizes Sertoli cells in mammals [46]. Human placenta and fetal membranes also express and co-localize AMH and AMHRII. Although no sex-related difference was found in their gene expression in both placenta and fetal membranes, intense staining for AMH in male fetal membranes supports AMH as a sex-specific hormone [47].

Sex differences in the placenta are also described: globally males have larger and heavier placenta, and birth weight/placental weight ratio than females [48,49,50]. The exact mechanism for these differences is unknown: some authors attribute it to sex differences in proliferation and metabolism at the earliest stages of blastocyst development [51,52,53], other reported males prioritize body growth, thus making them more vulnerable to sudden changes in fetal nutrition supply if they occur [48].

Moreover, placenta-associated pathologies are sexually divergent: chronic villitis and fetal thrombosis are more frequent in male placentas [54,55], while villous infarction is more common in females [54]. In addition, several sex differences are described in placenta gene expression, in hormonal asset, in immune response and hemodynamics [56,57,58,59,60,61,62], but this goes beyond the aims of this work and, therefore, will not be described in detail. As general examples, female placentas display higher expression of immune regulation genes, endocrine functions and placental growth [63,64], while male ones have more inflammatory profiles [65].

## 4. Fetal Programming and Sex

Fetal programming is the result of epigenetic changes that occurs in response to various stimuli that come from the environment that can affect the life and health of the baby even in adulthood [66]. According to Barker’s hypothesis (thrifty phenotype hypothesis) intrauterine growth retardation, low birth weight, premature birth and a low availability of nutrients during the prenatal stage may increase the risk of metabolic disorders, including Type II diabetes, hypertension, and coronary heart disease in middle age [66]. There are several factors involved in fetal programming: maternal smoking, malnutrition, stress, hormones, physical and psychological violence suffered by the mother and the fetus sex [9,10]. 

Fetal sex may affect the outcome of pregnancies: male sex is a risk factor for adverse pregnancy outcome, including preterm birth, premature rupture of membranes, gestational diabetes and macrosomia, motor and cognitive outcomes, and a lower likelihood of survival in intensive care [48,67,68].

Many mechanisms, processes, and systems that are activated during fetal developmental programming, such as gene expression, DNA methylation, telomere and mitochondrial biology, the sympathetic nervous system, the renin angiotensin system, oxidative stress, and inflammation, act in a sex-specific way [69,70,71,72].

Moreover, male fetuses are heavier than female ones at birth, and therefore they invest more energy in growth, adapting less to maternal conditions, while the female fetus conserves more energy during growth and this allows it to adapt better to maternal conditions in multiple ways [73]. The male fetus of mothers with severe asthma, for example, shows signs of impairment including intrauterine growth restriction (IUGR), preterm labor and stillbirth [74,75]. Females, on the other hand, adapt to the maternal condition of chronic asthma by reducing their growth, resulting in smaller but not non-IUGR. It has also been reported that the presence of a male fetus is associated with a maternal microvascular constriction in pre-eclamptic women. In pregnant women of a female fetus, maternal microvascular function was not significantly different between normotensive and hypertensive women [76,77].

Consequently, for example, male fetuses have less probability of survival than females when faced with adversity, as females react to adversities with a variety of strategies, avoiding the risk of early mortality or morbidity but paying the price of increased vulnerability expressed later in the development and during the lifetime [78]. Male fetuses in fact, invest resources in growth, and this strategy can contribute to their greater size at birth, but also to a relative poverty of resources to respond to subsequent exposure to stress and adversity. Because the male fetus has not conserved its resources, it has a limited ability to adapt to adversity and a greater risk of morbidity and mortality. By contrast, the female fetus does not invest so much in growth but conserves resources and adapts to maternal conditions in different ways [73,79,80].

## 5. Umbilical Cord

The umbilical cord contains two arteries that carry deoxygenated, nutrient-depleted blood away, and a vein, which carries oxygenated, nutrient-rich blood to the fetus [81,82]. 

The umbilical artery is made up of two main layers: an outer layer of muscle cells is found in a circular fashion and an inner layer with more irregularly available cells. The smooth muscle cells of the layer are poorly differentiated, containing only myofilaments [82].

Moreover, the umbilical cord contains WJ, a gelatinous substance made largely from mucopolysaccharides, which protects the blood vessels inside. WJ is enveloped in amniotic epithelium or, at the fetal end, a Malpighian keratinized epithelium, and it is a tissue that is active metabolically, involved in fluid exchange between umbilical vessels and amniotic fluid [83]. WJ is the primitive connective tissue of the human umbilical cord, described for the first time by Thomas Wharton in 1656 [84]. Subsequently, research efforts have attempted to optimize the isolation and differentiation of these cells from WJ [85,86].

An umbilical abnormality is represented by the presence of a single umbilical artery (SUA), a malformation that occurs when only one artery instead of two is present. In most cases, the baby is completely normal and healthy, but in a small percentage of babies the presence of a 2-vessel cord could indicate the presence of other abnormalities, sometimes life-threatening [87]. In fact, this condition may be associated with fetal growth restriction and increased perinatal mortality [87,88,89] and other birth defects, such as spina bifida associated with hydrocephalus [90]. Several studies reported that SUA is more common in female than male babies and is associated with multiparity and advanced maternal age [88,91,92,93]. Only one study describes a greater frequency of SUA in male than in female babies with an association with multiparity and advanced maternal age [90].

Moreover, the umbilical coiling index (number of 360-degree spiral course of umbilical vessels, and predictor of pregnancy outcome and risk of low birth weight [94]) is reported to be significantly higher in female than in male newborns, due to higher number of cord coils, without any difference in cord length [95].

Morphology of the umbilical cord may be dependent on the mother’s condition during pregnancy. It has been reported that in pre-eclamptic women there was an increase in the total area of the vessel, the total area of the vein, the total luminal area of the vein and the thickness of the wall of the arteries; the jelly area and the thickness of the vein wall decreased compared to the disease-free group [96]. In smoking pregnant women, umbilical arteries shows a thicker endothelial tissue with a different cell displacement [97]. Moreover, umbilical cord from smokers suffering of IUGR display a higher content of WJ and a decreased area of the umbilical vessels in comparison with healthy samples [98]. In addition, A recent systematic review and meta-analysis aimed to assess the association of fetal sex with multiple maternal complications; on 74 studies selected, the occurrence of pregnancy complications differed according to fetal sex with a higher cardiovascular and metabolic load for the mother in the presence of a male fetus. All pregnancy complications (i.e., gestational hypertension, total pre-eclampsia, eclampsia, placental abruption, and post-partum hemorrhage) tended to be associated with male fetal sex, except for preterm pre-eclampsia, which was more associated with female fetal sex [99].

## 6. Sex Differences in Cord Blood Cells, Plasma and Serum

Numerous biomarkers are influenced by sex [2,100,101,102,103,104,105], and this is true also for cord blood, plasma and serum biomarkers (Table 1).

It has been reported that in serum from vein of male umbilical cord the concentration of total testosterone, free testosterone, and estradiol, and inhibin (an inhibitor of FSH) are higher than in females [106]. Moreover, the authors reported that dehydroepiandrosterone sulfate from arterial serum was higher than that from veins only in female samples [106], confirming that biochemical parameters may also depend on the site of blood sampling in a sex-specific way [5].

Umbilical cord concentrations of cortisol and corticosterone are higher in the female fetus [107,108], while growth hormone (GH) is higher in male cords [109].

Some authors observed that cord plasma insulin and C-peptide concentrations were higher in female fetuses than in male ones, assuming a possible insulin resistance in females [110,111]. Moreover, umbilical cord concentrations of leptin are significantly increased in female fetuses, and it is associated with a higher placental weight only in females [112].

Some inflammatory and oxidative stress markers display a sexual dimorphism. For example, in plasma collected from premature twins the levels of 15-F(2t)-isoprostane (prostaglandin-like compounds formed in vivo from the free radical-catalyzed peroxidation of essential fatty acids) was higher in premature males than in premature females, and this sex differences in vulnerability to lipid oxidants that occurs early in life could represent a biological mechanism contributing to sex disparity later in life [113]. Moreover, glutathione levels are higher in segments of male umbilical cord vein perfused with tert-butylhydroperoxide, an inducer of oxidative stress [114]. It has been reported also that high prenatal exposure to carbon monoxide as air pollutant is associated with a significant reduction in cord blood mononuclear cell mitochondrial DNA copies, an oxidative stress biomarker, only in males [115].

Differences in DNA methylation are reported: cord blood from females have twice the number of methylated CpGs sites than males, which are associated with gene expression differences in many tissues such as brain, musculoskeletal, endocrinological and genitourinary [116].

Sex significantly affects cord blood complete blood count: male neonates have higher red blood cells, hemoglobin, hematocrit, and mean corpuscular hemoglobin concentration, and lower mean corpuscular volume, platelet and white blood cells counts than female neonates [117,118]. They also have higher lymphocyte, monocyte, eosinophil, basophil and lower neutrophil, metamyelocyte, myelocyte, and promyelocyte ratios than females [118]. Cord blood mononuclear cells are reported to be lower in males than in females. Moreover, CD34+ progenitor cells from male cord blood are significantly higher than those of female ones, and have higher capacity to produce colonies [119,120]. Moreover, a lower proportion of ILC2s (a type of innate lymphoid cell) are present in cord blood of human female neonates compared to males [121].

Fadini and colleagues reported that female newborns had a higher number of CD34+KDR+ endothelial progenitor cells than males [122]. Interestingly, progenitor cells from male and female cord blood display different gene expression: 1205 genes are upregulated in males and are related with sister chromatid segregation, chromosome segregation, neural precursor cell proliferation, mitotic sister chromatid segregation and positive regulation of cell proliferation. By contrast, 1313 genes related to platelet activation, response to wounding, wound healing, cell activation and blood coagulation are upregulated in females [119]. These differences are also associated with sexually different signaling pathway: males have high expression of CD5, CD8B, CD20, CD21, CD24, CD126, CD127 and interleukin-7, mainly associated with lymphocyte function, while high expression of CD41, CD42, CD61 and thrombopoietin, associated with platelet function characterized females [119].

**Table 1 life-11-00052-t001:** Sex differences in cord blood, plasma and serum.

Parameters	Source	M vs. F	Comments	Reference
**Total and free testosterone**	serum (venous)	M > F	Dehydroepiandrosterone sulfate from arterial serum > than that from vein only in F	[86]
**Estradiol**	serum (venous)	M > F		[86]
**Inhibin**	serum (venous)	M > F		[86]
**Cortisol and corticosterone**	serum (arterial and venous)	M < F		[107,108]
**Growth hormone**	serum (venous)	M > F		[109]
**Leptin**	serum (arterial and venous)	M < F		[92]
**Insulin and C**-**peptide**	plasma	M < F		[110,111]
**15-F**(**2t**)-**isoprostane**	plasma	M > F	Premature twins	[93]
**Gluthatione**	umbilical cord vein	M > F	Segments of umbilical cord vein perfused with tert-butylhydroperoxide	[94]
**Mononuclear cell mitochondrial DNA copies**	cord blood	M < F	After prenatal exposure to carbon monoxide as air pollutant	[95]
**DNA methylation**	cord blood	M < F	Number of methylated CpGs sites	[116]
**Red blood cells**	cord blood	M > F		[117,118]
**Hematocrit**	cord blood	M > F		[117,118]
**Hemoglobin**	cord blood	M > F		[117,118]
**Mean corpuscular hemoglobin concentration**	cord blood	M > F		[117,118]
**Mean corpuscular volume**	cord blood	M < F		[117,118]
**Platelets**	cord blood	M < F		[117,118]
**White blood cells**	cord blood	M < F	Lymphocyte, monocyte, eosinophil, basophil > M, neutrophil, metamyelocyte, myelocyte, and promyelocyte ratios > F	[118]
**CD34**+ **progenitor cells**	cord blood	M > F	M have higher capacity to produce colonies	[119,120]
**ILC2s**	cord blood	M > F		[121]
**CD34** + **KDR** + **progenitor cells**	cord blood	M < F		[122]

M = males; F = females.

## 7. Sex Differences in -HUVECs and HUAECs

HUVECs are a widely used in vitro model for the study of endothelium physiology and pathology [123,124]. Endothelial function and dysfunction display sex differences [125,126,127], but, although this aspect is clear, many authors still use HUVECs without reporting the donor sex. However, when male and female HUVECs were separately studied many differences emerged. Firstly, it is possible to observe male and female phenotypes: a higher rate of proliferation and migration, and higher levels of both the gene and protein for nitric oxide synthase 3 are observed in female cells than in male ones [15,128]. By contrast male HUVECs seems to have a higher degree of constitutive autophagy (an homeostatic mechanism, which maintain also normal cardiovascular function and morphology, through the lysosomal apparatus [129,130]): beclin-1 and the ratio LC3-II/LC3-I (the hallmark of the degree of autophagy activation), molecules involved in the different stages of autophagy, are significantly higher in male HUVECs, while some autophagy regulators, such as the mammalian target of rapamycin (mTOR) and the protein kinase B (AKT) are similar [15]. 

Moreover, HUVECs from males resulted in being more apoptotic than female ones after serum starvation, while no significant sex differences were observed in the percentage of necrotic cells [131].

The gene and protein expression of estrogen receptors (ERα, ERβ and GPER) and androgen receptor (AR) are not different between sexes [15], but no consensus exist on this aspect: some authors report that male and female HUVECs do not express ERα, while ERβ and AR expression is similar [132], while others show that HUVECs of unknown sex lack ERα and progesterone receptor (PR) type B (PRB) but express ERβ and PRA [133].

The vasoconstrictor thrombin is more efficient in female HUVECs than in male ones, in stimulating prostacyclin and prostaglandin E2 synthesis [134], and RLIP76, a Ral effector GTPase-activating protein, significantly altered the percentage of apoptosis only in female cells [135].

Many sex differences are reported with respect to tolerance to hypoxia, mRNA expression, and responses to shear stress [132,136,137]. Lorenz et al. demonstrate that 70 genes are differentially expressed between the sexes: female HUVECs have a larger levels of genes related to the immune response and some genes involved in metabolism (for example, leptin, insulin receptors and some apolipoproteins), and that they also have a greater capacity to form tubes and tolerate the stress of serum deprivation better than their male counterparts [136]. These results indicate that there are some sex differences in autosomal genes, common to both sexes, rather than through expression of sex chromosome genes or sex hormones [138], and these sex-associated differences in gene expression may strongly affect the risk, incidence, prevalence, severity and age-of-onset of many diseases [139,140].

Moreover, a higher number of genes are up- or down-regulated in female HUVECs than in male ones, after shear stress induction: vascular cell adhesion protein 1 expression is down-regulated almost 22 times in female HUVECs and only 3.5 times in male HUVECs [136]. Finally, unstimulated male HUVECs release more monocyte chemoattractant protein 1 (MCP1) and interleukin 8 than female HUVECs [141]. A brief exposure to tert-butylhydroperoxide induces a higher mortality in male HUVECs than in female ones [142]. More recently it has been reported that male and female HUVECs diverged in their secretome: 20 proteins (mostly related to responses to stress, cytokine stimulus, and apoptosis) are more abundant in male cells, while only 3 proteins are more present in the female secretome (retinal dehydrogenase 1, 6-phosphogluconate dehydrogenase decarboxylating, and transitional endoplasmic reticulum ATPase) [131].

Using twin pairs to study sex differences in transcriptome, Hartaman and colleagues observed that HUVECs from twins of different sexes (male/female) had greater differences in their transcriptome (2528 differentially expressed gene versus 79 in the comparison boy–boy versus girl–girl twins) than HUVECs from twins of the same sex (both male/male and female/female pairs). Females in boy/girl twins showed higher activation of endothelial pathways (endothelial to mesenchymal transition, hypoxia and nuclear factor-kappa B signaling), while males had significantly higher expression of Myc targets, oxidative phosphorylation and mTOR signaling [143]. Interestingly, some of the detected sex differences were maintained throughout life, confirming that sexual dimorphism starts in utero [143].

As regards HUAECs, no sex-related differences have been reported at present and to our knowledge, probably because they are a less used source of endothelial cells.

## 8. Sex Differences in HUASMCs

Few results are available about the influence of sex on HUASMCs. Cells from male and female neonates display sexual dimorphism in ERβ expression, with ERβ being more highly expressed in male-derived cells, while ERα is similarly expressed in both sexes [23]. ERα is also localized in HUASMCs starved for 5 days to allow for ERα up-regulation, and ERα is more highly expressed on average in HUASMCs from female donors than in HUASMCs from male donors [144].

Constitutive autophagy is similar between male and female HUASMCs, but they respond differently to pharmacological stimulations: serum starvation and rapamycin treatment (immunosuppressant and anticancer agent acting as a selective inhibitor of mTOR protein kinase, a pleiotropic agent in nutrient detection and signaling) [145] promote authophagy in both sexes, but especially in female cells increasing the LC3II/I ratio and decreasing the phosphorylation of the autophagic regulator mTOR. In addition, verapamil is able to increase LC3II/I ratio similarly in male and female HUASMCs, but it has a sex-specific effect in beclin-1 expression, indicating that treatments may activate sexually different signaling pathways in male and female HUASMCs autophagic process [23].

To the best of our knowledge, no further data on sex differences are available for a HUASMCs model.

However, it has been reported that proliferation of HUASMCs is inhibited by estrogen or progesterone at physiological concentrations, and this correlates to the inhibition of MAPK and MEK activities, while testosterone has no effect; unfortunately no indication of the infant’s sex is provided [146].

## 9. Sex Differences in WJ Cells

The WJ cell population expresses the characteristic phenotype of MSCs, exhibiting plastic adhesion and the expression of CD90, CD73 and CD105 [147]. Moreover, there are no safe data on the expression of estrogen and androgen receptors but only some reports regarding the effect of estradiol or testosterone on these cells [148,149,150].

Amniotic fluid-derived MSCs and WJ-MSCs, among the various tissues sources of stem cells, represent a promising cell population due to their high pluripotency, and WJ-MSCs have also attracted interest for their banking and transplantation capacities [151,152].

Balzano et al. [153], recently, disclosed novel biomedical implications in WJ-MSCs related to the sex of the donor, thus providing additional cues to exploit their regenerative potential in allogenic transplantation. They reported significantly higher gene expression of octamer-binding transcription factor 4 (OCT4), pluripotency gene, and the DNA-methyltransferase epigenetic modulator gene (DNMT1) in males than in females, while no sex differences have been detected in the expression levels of other stemness-regulating genes such as SOX2, NANOG, and C-MYC [153]. In a later study, the authors suggest that sex may affect the potential and efficiency of WJ-MSCs differentiation and autophagy: no significant differences between males and females were observed for miR-145-5p (target: OCT4 gene), and miR-185-3p (target: DNMT1 gene), while miR-148a-3p (target: OCT4 gene) was significantly lower in males. In addition, the autophagic marker LC3II/I ratio was higher in female cells than in male ones, indicating a higher constitutive autophagy in female cells [154].

Another study showed that baculoviral IAP repeat-containing protein 2 (BIRC2) and BIRC3 genes, which inhibit apoptosis by interfering with the activation of caspases, are higher, although not significantly, in WJ stem cells from male newborns, indicating, perhaps a sex difference in the sensitivity to apoptosis [155].

## 10. Conclusions

Understanding the sex-specific mechanisms underlying susceptibility to future diseases could lead to sex-specific preventive interventions during early childhood. From this overview of the available literature, it emerges that the sex variable is still often forgotten in experimental models. In fact, for some cell types, which may be important for understanding sex differences in the pathophysiology of the cardiovascular system, such as the one we have analyzed, there are no data. The knowledge of sex differences is fundamental to the improvement of therapeutic response, at least for cardiovascular diseases in both men and women, as they are the main cause of mortality and morbidity for both sexes. In the era of personalized medicine, it is clear that animals, organs, cells and organelles of male and female origin should be used for drug screening and in diagnostic procedures in order to provide sex-based medicine that could lead to new therapy approaches and strategies, increasing the adequacy and safety of therapy. The inclusion of XX cells and female animals in experiments and the analysis of data by sex can contribute to solving, at least in part, the problem of irreproducibility observed in preclinical biomedical research, paying particular attention to methodological problems [3]. Therefore, our key message is that it is no longer reasonable to ignore methodological issues in sex-specific research, because only awareness of these issues can lead to innovations. It is no longer sufficient to simply compare males and females on a range of health indicators, but there is a pressing need to use more sophisticated experimental designs, redefine old methods and develop new ones to produce new measures to study the influence of sex on health. A multilevel approach including molecular and cellular studies, the use of appropriate animal models, and well-designed human studies is required. In human research, experimental manipulation of prenatal stress and the intrauterine environment and access to many of the target tissues of interest, particularly in fetal life, are difficult to achieve. Therefore, for future research purposes, having simple experimental models available becomes primary for the comprehension of mechanisms of fetal programming.

## Figures and Tables

**Figure 1 life-11-00052-f001:**
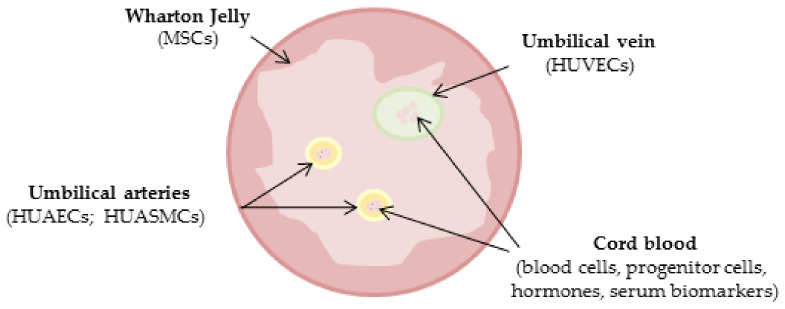
Schematic representation of human cord compartments.

## Data Availability

Data sharing not applicable.

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
