# Peer review of "Human Umbilical Cord: Information Mine in Sex-Specific Medicine"

_life, 2021, doi:10.3390/life11010052_

Round 1

Reviewer 1 Report

This review provides an overview that the sex needs to be considered in all aspects of life science research. Nevertheless, in some paragraphs it lacks sophisticated examples that support the view of the authors and that give readers deeper insight to the topic. It provides common information that the sex is important but no further information. 

It needs English editing (grammar and wording) and scientific language should be used (not e.g.  it seems, boys and girls, baby, …). Questionable wording and unclear paragraphs are highlighted but not limited to.

Also, if male embryos are more prone to miscarriage, should not there be much more males on earth. It is known that the bodies of males and females are different and that differentiation already starts during development. It would be of interest to provide examples what transcription factors are upregulated/downregulated that initiate the differences in development. 

L33: Examples of where it is important  should be given. 

L45: Cells should be specified. From cell lines used e.g. cancer research that are in culture for decades the gender is know and basic biochemical pathways may not differ between sexes. 

L46 and 49 are redundant.

L89: Write out abbreviation when they appear first.

L94: Is there a difference between RNA expression and IHC?

L122: Explain better. L123: A little bit more details, examples with references will help. 

L123-L126: Without reading the reference this paragraph has no value beside saying the females have better mechanisms to counteract any vulnerabilities during pregnancy. These mechanisms should be mentioned and discussed.

e.g. L141-L144: The reader may imagine what is meant but this is not clear and needs to be rewritten.

L230: Please elaborate on the sex differences in autosomal genes.

L241-244: This paragraph is not clear. Please rewrite.

L260: Is this cited study the only one describing sex differences in pharmacological studies. Please clarify and elaborate on the differences – rapamycin relates to mTOR.

L304: What specific preventive interventions are meant. please elaborate. Treating a male just because it is a male to prevent male related diseases?

Reviewer 2 Report

In this review article, the authors summarize the knowledge of sex differences observed in the human umbilical cord and highlight the importance of increasing awareness of sex differences in research and personalized medicine. This would be helpful for readers to notice the sex-related factors in their respective fields.
However, since information is scrappy and not well organized, it is difficult to understand how they relate to each other. There are several concerns that should be addressed. In my opinion, it requires major revisions before ready for publication.

Concerns:
(1)
Lines 68-70, ‘Afterwards, ... phenotypic sex[27];
=> It would be recommended to add some information on which hormones are produced in which cells of the differentiated male gonads. Especially, AMH produced by the Sertoli cells plays key roles in regression of the Mullerian duct, a female reproductive tract, in the male embryo.
Associated with this, lines 88-95 may need to be revised.

(2)
Lines 71-74, ‘In human pluripotent …areas of male fetuses.’;
=> The descriptions about the X inactivation is not correct or not sufficient. The study of ref-26 demonstrated that X-chromosome-dosage-compensation is not required for ESC differentiation. Hence, the sentence ‘Within this context …’ does not make sense. These sentences should be revised.
Some references regarding X-inactivation:
https://dev.biologists.org/content/147/1/dev183095
https://pubmed.ncbi.nlm.nih.gov/16285873/

(3)
Lines 106-126, the section of ‘Fetal programming and sex’;
=> It is not clearly explained what fetal programming is. Please add explanations of Barker’s hypothesis.

(4)
Lines 158-155, ‘Morphology of umbilical … samples [79]’;
=> It is not clear how these sentences are associated with other parts of the manuscript. It looks not associated with sex differences at all.

(5)
Lines 171-173, ‘As example, … than females’;
=> Please add a description of which stage/age such difference is observed.

(6)
Line 237, ‘in thei secretome’;
=> Typo?

(7)
Lines 241-247, ‘Using twin pair … throughout life’;
=> Please add information on which organs/cells showed such differences.

(8)
Line 262, ‘increasing L3II/I ratio’;
=> What does the L3II/I ratio indicate?

(9)
Line 263, ‘the autophagic regulator…’;
=> Please add information on how autophagy is involved in the development and/or functions of the human umbilical artery smooth muscle cells.

(10)
Line 278, ‘stemness gene regulation during early during early male germ cells’;
=> It is not clear why male germ cells are highlighted here. Please add explanations of how WJ cells are correlated with male germ cells.

(11)
‘gender differences’, ‘sex differences’, and ‘sex-gender’;
=> What is the different between ‘gender differences’ and ‘sex differences’? What does ‘sex-gender’ mean in the manuscript?

(12)
It would be recommended to add a section of ‘Abbreviations’.

Reviewer 3 Report

The present review manuscript by Campesi et al introduced and discussed the recent findings and advances regarding sex differences in the umbilical cord as well as fetal programming. Overall, it is a well organized and comprehensively described review article that would significantly contribute to the field of sexual dimorphism. I only have two suggestions as following.

  1. It would be better if the authors more focus on the sex differences in umbilical cord rather than fetal programming, especially in the Introduction and Conclusion.
  2. Adding a paragraph about the sex difference in umbilical cord and fetal programming in disease conditions, such as preeclampsia, gestational hypertension and gestational diabetes, may further promote the clinical relevance and significance of the present review.

Author Response

I wish to thank the referee for the comment and the suggestions that help us to improve the manuscript. Below are the answers to the required revisions (in blue in the text).

REVIEWER  3

The present review manuscript by Campesi et al introduced and discussed the recent findings and advances regarding sex differences in the umbilical cord as well as fetal programming. Overall, it is a well-organized and comprehensively described review article that would significantly contribute to the field of sexual dimorphism. I only have two suggestions as following.

  1. It would be better if the authors more focus on the sex differences in umbilical cord rather than fetal programming, especially in the Introduction and Conclusion. We think that sex differences observed in umbilical cord are strictly connected to the fetal programming as they may predispose to future diseases. the space dedicated to this aspect is, in our opinion, non-invasive and does not detract from the description of the sex differences reported in the umbilical cord compartments.
  2. Adding a paragraph about the sex difference in umbilical cord and fetal programming in disease conditions, such as preeclampsia, gestational hypertension and gestational diabetes, may further promote the clinical relevance and significance of the present review.

The following information has been added in the paragraph 5.: “ Morphology of umbilical cord may be dependent on the mother's condition during pregnancy. It has been reported that in pre-eclamptic women there was an increase in the total area of ​​the vessel, the total area of ​​the vein, the total luminal area of ​​the vein and the thickness of the wall of the arteries; the jelly area and the thickness of the vein wall decreased compared to the disease-free group [96]. In smoking pregnant women umbilical arteries shows a thicker endothelial tissue with a different cell displacement [97]. Moreover, umbilical cord from smokers suffering of IUGR display a higher content of Wharton’s jelly and a decreased area of the umbilical vessels in comparison with healthy samples [98]. In addition, A recent systematic review and meta-analysis aimed to assess the association of fetal sex with multiple maternal complications; on 74 studies selected, the occurrence of pregnancy complications differed according to fetal sex with a higher cardiovascular and metabolic load for the mother in the presence of a male fetus. All pregnancy complications (i.e. gestational hypertension, total pre-eclampsia, eclampsia, placental abruption, and post-partum hemorrhage) tended to be associated with male fetal sex, except for preterm pre-eclampsia, which was more associated with female fetal sex [99]”

Round 2

Reviewer 1 Report

While the manuscript hast slightly improved, it still needs major English editing, contains typos that make it difficult to read, lacks consistency, and contains flaws and non-clear statements.

Some examples:

L27: remove dot before innovative.
L33: missing comma between items in enumerations.
L76: space after The.

L69: If there are two stages it would be better to use first and second instead of first and next!

L75: silencing is often not complete. What is the outcome of that. Please discuss.
L90: It should read males not females.

consistency with abbreviations:
L81: fibroblast growth factor 9 should be followed by FGF9.
L98: transforming growth factor-beta should be abbreviated as TGF-beta, not TGF.
L109: SRY and SRY-box SOX9 should read SRY and SOX9

L106 and L107: include (males) and (females) otherwise you could conclude that in the same organism some cells are expressing SRY while others don't.

L114 and L141: males are heavier at birth and have larger and heavier placentas compared to females. Why do males have larger and heavier birth weight/placental weigth ratio. difficult to understand and please explane
since larger weight/size and lager placenta leads to same ratio.

Figure 1.
Please indicate the arteries with arrows that point the same way to the arteries. Otherwise it may be confusing. Same for umilical vein and Wharton Jelly arrows.

Please remove doi:doi in references.

Author Response

I wish to thank the referee for the comment and the suggestions that help us to improve the manuscript. Below are the answers to the required revisions (in blue in the text).

REVIEWER 1

While the manuscript hast slightly improved, it still needs major English editing, contains typos that make it difficult to read, lacks consistency, and contains flaws and non-clear statements.

Some examples:

L27: remove dot before innovative; L33: missing comma between items in enumerations; L76: space after The.; L69: If there are two stages it would be better to use first and second instead of first and next!: We thank the referee for her/his suggestion, the text has been entirely revised to check for any grammatical errors

L75: silencing is often not complete. What is the outcome of that. Please discuss. As requested, the concept has been developed and discussed

L90: It should read males not females. Crown-rump length and biparietal diameter are larger in males than in females as confirmed by the cited references.

Consistency with abbreviations: L81: fibroblast growth factor 9 should be followed by FGF9;  L98: transforming growth factor-beta should be abbreviated as TGF-beta, not TGF;  L109: SRY and SRY-box SOX9 should read SRY and SOX9. We apologise for the oversight, abbreviations are now corrected and reported in the list.

L106 and L107: include (males) and (females) otherwise you could conclude that in the same organism some cells are expressing SRY while others don't. The sentence has been rephrased to a better comprehension

L114 and L141: males are heavier at birth and have larger and heavier placentas compared to females. Why do males have larger and heavier birth weight/placental weight ratio. difficult to understand and please explain since larger weight/size and lager placenta leads to same ratio. The following sentence has been added in the section: “The exact mechanism for these differences is unknown: some authors attribute it to sex differences in proliferation and metabolism at the earliest stages of blastocyst development [51-53], other reported males prioritize body growth, thus making them more vulnerable to sudden changes in fetal nutrition supply if they occur [48]”.

Moreover in the revised version we have already reported that “Male fetuses, in fact, invest resources in growth, and this strategy can contribute to their greater size at birth, but also to a relative poverty of resources to respond to subsequent exposure to stress and adversity. Because the male fetus has not conserved its resources, it has a limited ability to adapt to adversity and a greater risk of morbidity and mortality. On the contrary, the female fetus does not invest so much in growth but conserves resources and adapts to maternal conditions in different ways”.

Figure 1. Please indicate the arteries with arrows that point the same way to the arteries. Otherwise it may be confusing. Same for umbilical vein and Wharton Jelly arrows. Done as requested

Please remove doi:doi in references. Done as requested

Reviewer 2 Report

All concerns by this reviewer have been appropriately addressed and answered. The revisions improved the manuscript and will help readers’ understanding. In my opinion, it looks acceptable for publication.

Regarding the revised sentences, there are a couple of recommendations to be addressed before publication.

(1) Lines 81-83, ‘Furthermore, Colvin et al. 2001… as males [31].’;
=> It is not clear how these sentences relate with lines 69-80. FGF9 and SOX9 are autosomal genes. Further, the reference of ‘Colvin et al. 2001’ is not listed in the References section.

(2) Line 111, ‘Human placenta and fetal membranes express and …’;
=> ‘also’?, ‘In addition to reproductive organs’?

(3) Lines 183-186,
=> ‘)’ is missing at somewhere.

(4) Table 1, column of ‘M vs F’;
=> It is not clear what ‘>’ symbol indicates. Does ‘> M’ mean that the level is lower in males than females? If so, it is better to indicate as ‘M < F’.

Author Response

I wish to thank the referee for the comment and the suggestions that help us to improve the manuscript. Below are the answers to the required revisions (in blue in the text).

REVIEWER 2

All concerns by this reviewer have been appropriately addressed and answered. The revisions improved the manuscript and will help readers’ understanding. In my opinion, it looks acceptable for publication.

Regarding the revised sentences, there are a couple of recommendations to be addressed before publication.

(1) Lines 81-83, ‘Furthermore, Colvin et al. 2001… as males [31].’; => It is not clear how these sentences relate with lines 69-80. FGF9 and SOX9 are autosomal genes. Further, the reference of ‘Colvin et al. 2001’ is not listed in the References section. The sentence has been rephrased to a better comprehension

(2) Line 111, ‘Human placenta and fetal membranes express and …’;=> ‘also’?, ‘In addition to reproductive organs’? The text has been corrected

(3) Lines 183-186,=> ‘)’ is missing at somewhere. The text has been corrected

(4) Table 1, column of ‘M vs F’; => It is not clear what ‘>’ symbol indicates. Does ‘> M’ mean that the level is lower in males than females? If so, it is better to indicate as ‘M < F’. We thank the referee for her/his suggestion, symbols in Table have been corrected indicating M > F and M< F to a better comprehension